# Study on Antibacterial Activity and Structure of Chemically Modified Lysozyme

**DOI:** 10.3390/molecules28010095

**Published:** 2022-12-22

**Authors:** Sheng-Wei Wang, Tian-Yi Wang

**Affiliations:** School of Marine Science and Technology, Harbin Institute of Technology, Weihai 264209, China

**Keywords:** hen egg white lysozyme, chemical modification, antibacterial activity, secondary structure

## Abstract

Lysozyme is a natural protein with a good bacteriostatic effect, but its poor inhibition of Gram-negative bacteria limits its development potential as a natural preservative. Therefore, the modification of natural lysozyme to expand the antimicrobial spectrum become the focus of lysozyme study. Egg white lysozyme has low cost, rich content in nature, is easy to obtain, strong stability, and high enzyme activity, so it can be applied in the modification of lysozyme. Egg white lysozyme was modified by chemical methods using organic acids. Caffeic acid and p-coumaric acid in organic acids were used as modifiers, and 1-Ethyl-3- (3-dimethylaminopropyl) carbodiimide hydrochloride and N-hydroxy succinimide were used as dehydration condensation agents during modification. A certain degree of modified lysozyme was obtained through appropriate modification conditions. The antibacterial properties and structure of the obtained two organic acid-modified lysozymes were compared with natural enzymes. The results showed that compared with the native enzyme, the activity of modified lysozyme decreased, but the inhibitory effect on Gram-negative bacteria was enhanced. The minimum inhibitory concentrations of caffeic acid-modified enzyme and p-coumaric acid-modified enzyme on *Escherichia coli* and *Pseudomonas aeruginosa* were 0.5 mg/mL and 0.75 mg/mL, respectively. However, the antibacterial ability of modified lysozyme to Gram-positive bacteria was lower than that of the natural enzyme. The minimum inhibitory concentration of caffeic acid-modified enzyme and p-coumaric acid-modified enzyme to *Staphylococcus aureus* and *Bacillus subtilis* was 1.25 mg/mL. The peak fitting results of the amide-I band absorption peak in the infrared spectroscopy showed that the content of the secondary structure of the two modified enzymes obtained after modification was different from that of natural enzymes. In the study, two organic acids were used to modify egg white lysozyme, which enhanced the enzyme’s inhibition of Gram-negative bacteria, and analyzed the mechanisms for the change in the enzyme’s antibacterial ability from the perspective of the structural change of the modified enzyme, providing a new idea for lysozyme modification.

## 1. Introduction

Lysozyme is a kind of naturally monomeric protein, which has a good antibacterial effect and widely exists in nature, and there are type C, type I, type G, and other types of lysozyme [1]. Hen egg white lysozyme is C-type lysozyme which consists of 129 amino acids. It is a glycoside hydrolase with a molecular weight of about 14,400 and an isoelectric point of about 11 [2]. The chemical property of hen egg white lysozyme is very stable. When the pH value is in the range of 1.2–11.3, the structure is almost unchanged, and the stability to heat is also very strong in the acid environment [3]. Lysozyme has the functions of antibacterial, antiviral, scavenging necrotic tissue, accelerating wound repair and regeneration, and is widely used in the food and pharmaceutical industries and it has been officially approved by many countries and organizations as a food preservative or preservative. However, natural lysozyme is a non-broad-spectrum antibacterial agent that mainly acts on Gram-positive bacteria (G+). Lysozyme achieves antibacterial activity by destroying the β-1,4 glycosidic bond between N-acetylglucosamine (NAG) and N-acetylmuramic acid (NAM) in the bacterial peptidoglycan layer [4]. Due to the high content of peptidoglycan in the cell wall of Gram-positive bacteria, lysozyme has a good antibacterial effect on Gram-positive bacteria. However, Gram-negative bacteria (G−) in vitro lipid outer membrane package and cell wall contains only a small amount of peptidoglycan, which to some extent affects the lysozyme on Gram-negative bacteria antibacterial effect, limiting its development potential as a natural preservative [5].

To change this limitation, Visalsok Touch et al. [6] used 2 mmol/L dithiothreitol to treat lysozyme for 0.5–4 h at pH 8.0 and 30 °C, which enhanced the bactericidal effect of lysozyme on Gram-negative bacteria. Ibrahim et al. [7,8] thought that structural modification of lysozyme with a membrane-fusing hydrophobic domain, such as saturated hydrophobic peptides or fatty acids, may facilitate the ability of lysozyme to access and move through the outer cell membrane and thus destroy Gram-negative bacteria, so Ibrahim added a hydrophobic peptide to the C-terminus of the lysozyme molecule, which increased the inhibitory effect of lysozyme on *Escherichia coli*, and Ibrahim et al. [9] also used long-chain fatty acids to modify natural lysozyme, so that the antibacterial ability of modified lysozyme to Gram-negative bacteria can be enhanced. In addition, Ibrahim et al. [10] also modified lysozyme with perillaldehyde, and synthesized Periliaedhyed-lysozyme1 polymer with the inhibitory ability to *Staphylococcus aureus* and *Escherichia coli* k12, Andreas and Berkop-Schnurch et al. [11] used cinnamic acid and caffeic acid to modify lysozyme, and found that lysozyme can bind to these aromatic organic acids through covalent bonds, so as to achieve the purpose of expanding the antibacterial spectrum of lysozyme. However, the modified lysozyme obtained by this modification method will reduce the inhibitory effect on Gram-positive bacteria while inhibiting.

In addition to the chemical modification of lysozyme, in recent years, the use of physical methods to modify lysozyme research has gradually increased. Under the conditions of high temperature, high pressure, vacuum, magnetic field, microwave, and so on, natural lysozyme is treated to a certain extent. Through the above physical treatment, the spatial conformation of lysozyme molecules has changed, thus changing some structural characteristics and biological functions of the enzyme. The modification method is called the physical modification of enzyme molecules. During the modification process, the covalent bond in the molecule does not change, but the secondary structure changes. Physical modification has little damage to the nutritional value of lysozyme and is non-toxic and time-saving. However, the range of modification by this method is relatively narrow. For some lysozymes, it has no significant modification effect. Therefore, the application of physical modification of lysozyme is limited. At present, the modification methods of lysozyme under high temperature and ultra-high-pressure conditions are commonly used to change the molecular structure of lysozyme. Hisham et al. [12,13] found that the heated lysozyme could effectively inhibit the growth of *Escherichia coli*, but the modification of lysozyme by heating did not affect the inhibition of lysozyme on Gram-positive bacteria to a large extent. After homogenizing natural lysozyme under 100 MPa high pressure, Lucci et al. [14] found that the inhibition of lysozyme on *Listeria monocytogenes* was significantly enhanced. At the same time, Alline et al. [15] proved that the antibacterial effect of lysozyme and the ultra-high pressure homogenization have a synergistic effect on the inhibition of *Lactobacillus acidophilus*.

In order to further explore the effect and change of lysozyme on antibacterial ability after chemical modification by organic aromatic acids, this study focused on the inhibitory effect of caffeic acid and p-coumaric acid-modified lysozyme on Gram-negative bacteria and the structural changes of the enzyme before and after modification.

## 2. Results and Discussion

### 2.1. Analysis of Column Chromatography Results

Separation and purification by Sephadex G-25 column, the elution curves are shown in Figure 1. Elution peaks 1,1′, first outflow, the corresponding component is a macromolecular substance, elution peaks 2,2′, after the outflow, the corresponding is a small molecule substance, were collected related components, measuring enzyme activity, can be obtained elution peaks 1,1′, with enzyme activity, is a mixture containing lysozyme, elution peaks 2,2′, does not have enzyme activity, is not involved in the reaction of free small molecule substances, mainly not involved in the reaction of free caffeic acid, p-coumaric acid, and other impurity molecules, so the collection of peaks 1,1′, with enzyme activity components freeze-dry.

### 2.2. Lytic Activity

The enzyme activity average values of natural lysozyme, caffeic acid-modified lysozyme, and p-coumaric acid-modified lysozyme were 20,583 U/mg, 14,243 U/mg, and 13,317 U/mg, respectively, which were measured by the experimental method and Equation (1) in Section 3.4.2. According to the mean of measured enzyme activity, the relative enzyme activity of lysozyme, which was measured with *Micrococcus lysodeikticus* as the substrate, is shown in Figure 2. In Figure 2, obviously, compared with the natural enzyme, the activity of the modified enzyme decreased. It was calculated that the activities of the caffeic acid-modified enzyme and p-coumaric acid-modified enzyme were 69.2% and 64.7% of the natural enzyme, respectively. The secondary structure test revealed that the content of each secondary structure of the enzyme changed before and after modification, indicating that the change in enzyme structure may be the primary explanation for the drop-in activity.

### 2.3. Antibacterial Effect

The results of the inhibition zone diameter of natural lysozyme and modified lysozyme on different test strains are shown in Figure 3. It can be seen from Figure 3 that the inhibitory effect of caffeic acid-modified enzyme and p-coumaric acid-modified enzyme on Gram-negative bacteria is stronger than that of Gram-positive bacteria. The inhibition zone diameters of the two modified enzymes against Gram-negative bacteria were greater than 13 mm, which was significantly larger than that of the natural enzyme. Therefore, the antibacterial effect of the modified enzyme on Gram-negative bacteria was significantly enhanced compared with the natural enzyme, and the antibacterial ability of the caffeic acid-modified enzyme was slightly stronger than that of the p-coumaric acid-modified enzyme. The results showed that the antibacterial effect of modified lysozyme on Gram-negative bacteria was stronger than that of natural lysozyme, but the inhibition ability of modified lysozyme on Gram-positive bacteria was weaker than that of natural lysozyme to some extent. The inhibitory effect of the modified enzyme on Gram-negative bacteria was enhanced, while the inhibitory effect on Gram-positive bacteria was weakened. The inhibition zone diameters of the two modified enzymes on Gram-positive bacteria were smaller than that of the natural enzyme. Compared with the antibacterial ability of natural lysozyme to Gram-negative bacteria and Gram-positive bacteria measured in the experiment, the antibacterial ability of the two modified enzymes to Gram-negative bacteria and Gram-positive bacteria has a similar change trend. It can be proved that the mechanisms for the change in the antibacterial ability of the two modified enzymes should be the same.

### 2.4. Minimum Inhibitory Concentration (MIC)

The MIC results of natural lysozyme and modified lysozyme to different test strains are shown in Table 1 and Table 2. By analyzing Table 1, the MIC of natural lysozyme and modified lysozyme in Table 2 can be obtained. It can be seen from Table 1 and Table 2 that the two modified lysozymes can effectively inhibit the growth and metabolism of Gram-negative bacteria to a certain extent. The MIC results of natural enzyme and modified enzyme against Gram-negative bacteria (*Escherichia coli* and *Pseudomonas aeruginosa*) were as follows: natural lysozyme 1.50 mg/mL, caffeic acid modified lysozyme 0.50 mg/mL, p-coumaric acid modified lysozyme 0.75 mg/mL, while the MIC of natural lysozyme against Gram-positive bacteria (*Staphylococcus aureus* and *Bacillus subtilis*) was 0.75 mg/mL, and the MIC results of modified lysozyme against two Gram-positive bacteria (*Staphylococcus aureus* and *Bacillus subtilis*) were all greater than 1.00 mg/mL. The determination results of MIC corresponded to the results of the antibacterial effect. Through the determination of MIC, it was finally determined that the antibacterial spectrum of modified lysozyme was expanded to a certain extent, and it was further confirmed that the reason for the change of the antibacterial ability of the two modified enzymes relative to the natural lysozyme should be identical, which facilitated further discussion and research.

### 2.5. Structure and Discussion

The amide-I band (1600–1700 cm^−1^) in the IR spectra of proteins is most sensitive to their secondary structure (α-helix, β-sheet, random coil, β-turn, etc.) [16,17]. Therefore, the amide-I region was evaluated to estimate the proportion of different secondary structures in natural lysozyme and modified lysozyme. In a D_2_O solution, the amide I band profile can be decomposed into several components. The wavenumber distributions of the four secondary structures (α-helix, β-sheet, random coil, β-turn) of proteins in the amide I band in the D_2_O solution are shown in Table 3. By peak fitting, the peak separation results are shown in Figure 4. According to the distribution range of the secondary structure in Table 3, the relative proportion of the four secondary structures in Table 4 is obtained by calculating the area ratio. The relative proportions of the secondary structure content of natural and modified enzymes are visually displayed in Figure 5.

From Table 4 and Figure 5, it can be seen that the content of each secondary structure of the modified enzyme has changed compared with the natural enzyme. Compared with the natural enzyme, the content of β-sheet increased after different acid modifications, from 28.10% to 38.87% and 32.62%, respectively. The content of the caffeic acid-modified enzyme and p-coumaric acid-modified enzyme random coil decreased compared with the native enzyme, and the decrease in the p-coumaric acid-modified enzyme was smaller than that of the caffeic acid-modified enzyme. The content of the α-helix of the caffeic acid-modified enzyme increased significantly compared with the native enzyme, but the content of the α-helix of the coumaric acid-modified enzyme decreased compared with the native enzyme. Compared with the content of natural enzyme β-turn, the content of caffeic acid-modified enzyme β-turn decreased significantly from 23.27% to 12.78%, while the content of p-coumaric acid-modified enzyme β-turn increased to 35.33%. By analyzing the relative content of the secondary structure of the natural enzyme and modified enzyme, it was deduced that the change of modified enzyme structure might be the reason for the change in enzyme activity and antibacterial activity. According to the measured changes in the secondary structure content, it can be inferred that it may be due to structural changes that the aromatic hydrophobic groups originally wrapped in the protein molecule are exposed. On the other hand, due to the structural characteristics of organic acids and their covalent binding with the enzyme molecules, during the binding process, the amino group inside the enzyme molecule is covalently bound to the carboxyl group inside the organic acid. The existence of the hydrophobic benzene ring contained in the organic acid may increase the hydrophobicity of its surface. It is speculated that the modification process enhances the surface hydrophobicity of the enzyme molecule so that the modified lysozyme can be compatible with the lipophilic outer membrane of Gram-negative bacteria compared with the natural lysozyme so that it is easier to contact and hydrolyze to the peptidoglycan wall of Gram-negative bacteria, thereby enhancing the antibacterial effect of the modified lysozyme on Gram-negative bacteria. It is speculated that it is also because the modification process increases the surface hydrophobicity of lysozyme, which reduces the inhibition of modified lysozyme on Gram-positive bacteria.

### 2.6. Comparison and Discussion

In the study, in addition to comparing the antibacterial ability of lysozyme chemically modified with organic acids to that of native lysozyme, the antibacterial ability of lysozyme chemically modified with organic acids must also be compared to that of lysozyme chemically modified with other chemicals. A comparison of similar works can better demonstrate the potential of this work for expanding lysozyme’s antibacterial ability. Table 5 summarizes the literature on the chemical modification of lysozyme and contrasts the consequences of these modifications. Indeed, a comparison of the contents of the literature reveals that the lytic activity of the chemically modified lysozymes was all reduced. However, the antibacterial ability against Gram-negative bacteria was enhanced to varying degrees, while the antibacterial ability against Gram-positive bacteria was reduced to varying degrees. In the study, the lytic activity values of caffeic acid-modified lysozyme and p-coumaric acid-modified lysozyme were lower than native lysozyme, but the antibacterial range was greater, demonstrating the potential use of caffeic acid and p-coumaric acid in lysozyme chemical modification through comparison.

## 3. Materials and Methods

### 3.1. Materials

Strains and reagents: *Staphylococcus aureus* ATCC43300, *Micrococcus lysodeikticus* ATCC4698, *Escherichia coli* ATCC25922, *Bacillus subtilis* ATCC6051, *Pseudomonas aeruginosa* ATCC27853, were provided by Biological Public Laboratory of Harbin Institute of Technology, Weihai. Hen egg white lysozyme (>20,000 U/mg, M_w_ = 14,400 Da), 1-Ethyl-3-(3-dimethylaminopropyl) carbodiimide hydrochloride (EDC), caffeic acid, p-coumaric acid and N-hydroxy succinimide (NHS) were all purchased from Shanghai Aladdin Biochemical Technology Company, and peptone, beef extract, agar powder, purchased from Sinopharm Chemical Reagent Company.

Main experimental equipment: ultra-clean worktable (SW-CJ-1FD, Lichen, China), high-pressure steam sterilization pot (DGL-35B, Lichen, China), constant temperature incubator (THZ-98AB, Yiheng, China), constant temperature water bath (HH-6S, Ny, China), freeze dryer (LCG-10, NANBEI, China), high-speed centrifuge (LC-LX-H165A, Lichen, China), UV-Visible spectrophotometer (UV-1800, Shimazu, Japan), Fourier Transform Infrared Spectrometer with Attenuated Total Reflectance cell (TENSOR Ⅱ, Bruker, Germany).

### 3.2. Preparation of Modified Lysozyme

30 mg caffeic acid and p-coumaric acid were dissolved in 5 mL 3 mol/L NaOH solution, adjusted to pH 7.5 with 5 mol/L HCl, and then 120 mg EDC and 80 mg NHS were added to each organic acid solution, respectively. After complete dissolution, the solution was placed at room temperature of 25 °C for 40 min. Then 60 mg lysozyme was added to the solution. In this reaction system, the mass ratio of organic acid to lysozyme was 1:2, and the mixture was stirred in a water bath at 35 °C for 24 h. After the reaction was completed, the insoluble part of the system was removed by centrifugation (4000× *g*, 15 min), and the supernatant was retained for subsequent experiments.

### 3.3. Sephadex G-25 Column Chromatography

For purification of the modified lysozyme, 5 mL of the reaction solution sample was injected into the Sephadex G-25 column and eluted with 0.1 mol/L, pH 6.2 phosphate buffer (PBS). The flow rate was 0.8 mL/min, and the detection wavelength was 281 nm. Collection of peaks with lysozyme activity, freeze drying, removal of unreacted free caffeic acid, p-coumaric acid, and other impurity molecules.

### 3.4. Determination of Lysozyme Activity

#### 3.4.1. Preparation of *Micrococcus lysodeikticus* Suspension

*Micrococcus lysodeikticus* was inoculated in a liquid medium and cultured at 28 °C and 200 rpm for 24 h. Under the condition of 5000× *g*, 15 min, the bacterial solution obtained by expanding culture was centrifuged, the final bacterial precipitation was washed three times with sterile saline, and the final bacteria were frozen with sterile glycerol as a protective agent. When the lysozyme activity was measured, the 0.1 mol/L, pH 6.2 PBS was used to dilute it into a certain concentration of bacterial suspension so that the OD value at 450 nm was about 1.0.

#### 3.4.2. Determination of Enzyme Activity

5 mg lysozyme was dissolved in 0.1 mol/L pH 6.2 PBS so that the concentration of the enzyme solution to be tested was 1 mg/mL. After the substrate suspension and the enzyme solution to be tested were placed in a water bath at 25 °C for 30 min, the substrate suspension was taken out to determine its OD value under 450 nm, and then the enzyme solution to be tested was added and quickly shaken up. The OD_450_ value was measured once every 30 s for three consecutive measurements. At room temperature, the decrease in OD_450_ value per 30 s was 0.001 as an enzyme activity unit (25 °C, pH 6.2). The unit of enzyme activity per milligram (U/mg) can be expressed by Equation (1). The enzyme activity of the native enzyme and modified lysozyme was determined by this method. The highest enzyme activity was 100%, and the relative enzyme activity was the ratio of the measured enzyme activity to the highest enzyme activity.
(1)$$U/\mathrm{mg}\;=\frac{\left({\Delta \mathrm{OD}}_{450}/\;\min\right)}{\mathrm{mass}\;\mathrm{of}\;\mathrm{enzyme}\;\mathrm{sample}\;\left(\mathrm{mg}\right)}\times 10^{3}$$

### 3.5. Determination of Antibacterial Ability

#### 3.5.1. Preparation of Bacterial Suspension

On the super clean bench, a small amount of *Escherichia coli*, *Staphylococcus aureus*, *Bacillus subtilis*, and *Pseudomonas aeruginosa* were picked with inoculation ring and inoculated on sterilized culture media respectively and cultured at 28 °C for 24 h. Then the single colonies of various bacteria were picked with the inoculation ring and inoculated into the sterilized liquid medium. The liquid medium was placed in a constant temperature shaker at 28 °C for 24 h. The culture of each strain was diluted to 0.5 McFarland’s turbidity by McFarland’s turbidimetry, i.e. [25], the culture concentration was diluted to 1.5 × 10^8^ CFU/mL.

#### 3.5.2. Determination of Inhibition Zone

The 20 mg of natural hen egg white lysozyme, caffeic acid-modified lysozyme, and p-coumaric acid-modified lysozyme were weighed, respectively. In the preparation process of the sample solution, 1 mL 0.1 mol/L pH 6.2 phosphate buffer was used as the solvent so that the final concentration of the sample solution was 20 mg/mL. By using the filter paper method, each test bacterial solution (200 μL) that had been purified and cultured and diluted to 1.5 × 10^8^ CFU/mL was coated in a solid medium. After the bacterial solution on the surface of the medium was fully dried, the sterilized filter paper was placed on the surface of the medium, and 20 μL of the prepared enzyme sample solution was respectively added dropwise on the filter paper and cultured at a constant temperature of 28 °C for 24 h [26,27]. The diameter of the inhibition zone of each enzyme sample solution in the petri dish against different strains was measured.

### 3.6. Determination of Minimum Inhibitory Concentration (MIC)

15 mg of the native enzyme and modified enzyme were dissolved in 10 mL, 0.05 mol/L pH 6.2 phosphate buffer solution, respectively. In addition, the sample solutions with mass concentrations of 1.50 mg/mL, 1.25 mg/mL, 1.00 mg/mL, 0.75 mg/mL, 0.50 mg/mL, and 0.25 mg/mL were prepared by gradient dilution [27,28]. 200 μL, 1.5 × 10^8^ CFU/mL *Staphylococcus aureus*, *Bacillus subtilis*, *Escherichia coli*, and *Pseudomonas aeruginosa* were added to 200 μL different sample solutions respectively, and each concentration was repeated three times. In addition, incubated at 28 °C constant temperature incubator for 2 h, different mixtures were diluted by 1:1000 with sterilized deionized water, and 200 μL of the diluted solution was coated on the medium and cultured at 28 °C for 24 h to observe the growth of colonies. The sample solution was replaced by 200 μL, 0.05 mol/L pH 6.2 phosphate buffer as blank control. The colony growth on the plate was observed, and the concentration of the sample in the sterile growth plate was MIC.

### 3.7. Fourier-Transform Infrared (FTIR) Spectroscopy and Protein Secondary Structure

Infrared spectra of enzymes were recorded using an FTIR spectrometer equipped with Attenuated Total Reflectance (ATR) accessory. D_2_O has no interference absorption in the amide I band (1600–1700 cm^−1^), and the amide I band (1600–1700 cm^−1^) is an important wavenumber segment for the secondary structure information of the protein, so an appropriate amount of natural hen egg white lysozyme, caffeic acid modified lysozyme and p-coumaric acid modified lysozyme was dissolved in D_2_O to prepare a D_2_O solution and sampled for ATR-FTIR scanning [18,19,29]. ATR-FTIR Set parameters: Spectral resolution 4 cm^−1^, Wavenumber scanning range 4000–400 cm^−1^, Scan times 32. After obtaining the infrared spectrum, Peak Fit v4.12 software (Systat Software, Inc., San Jose, CA, USA) was used to analyze the l600–1700 cm^−1^ band belonging to the characteristic peak of the amide I band in the infrared spectrum. Firstly, the baseline was corrected, and then the Gaussian method was used for deconvolution, and then the second derivative method was used to fit, and the multiple fitting was performed until it coincided with the original spectrum [16,30]. After the fitting was completed, the characteristic peak position of each conformation and the ratio of the area of each characteristic peak to the total area were obtained, and the proportion of each conformation in the protein secondary structure was obtained.

## 4. Conclusions

Compared with natural enzymes, although the enzyme activity of hen egg white lysozyme modified by caffeic acid and p-coumaric acid decreased, the antibacterial effect on Gram-negative bacteria was enhanced. The minimum inhibitory concentrations of caffeic acid and p-coumaric acid modified enzyme against Gram-negative bacteria (*Escherichia coli* and *Pseudomonas aeruginosa*) were 0.5 mg/mL and 0.75 mg/mL, respectively, while the inhibitory effect on Gram-positive bacteria was weakened. The minimum inhibitory concentrations of the two modified enzymes against *Staphylococcus aureus* and *Bacillus subtilis* were greater than those of natural enzymes. Compared with the natural enzyme, the content of each secondary structure of the modified enzyme changed to some extent. The change in the structure of this enzyme is the reason for the change in enzyme activity and antibacterial activity. This study provided guidance and expanded ideas for the application of natural hen egg white lysozyme modification to expand its antibacterial spectrum and enhance the antibacterial ability of lysozyme against Gram-negative bacteria in the pharmaceutical and food industry.

## Figures and Tables

**Figure 1 molecules-28-00095-f001:**
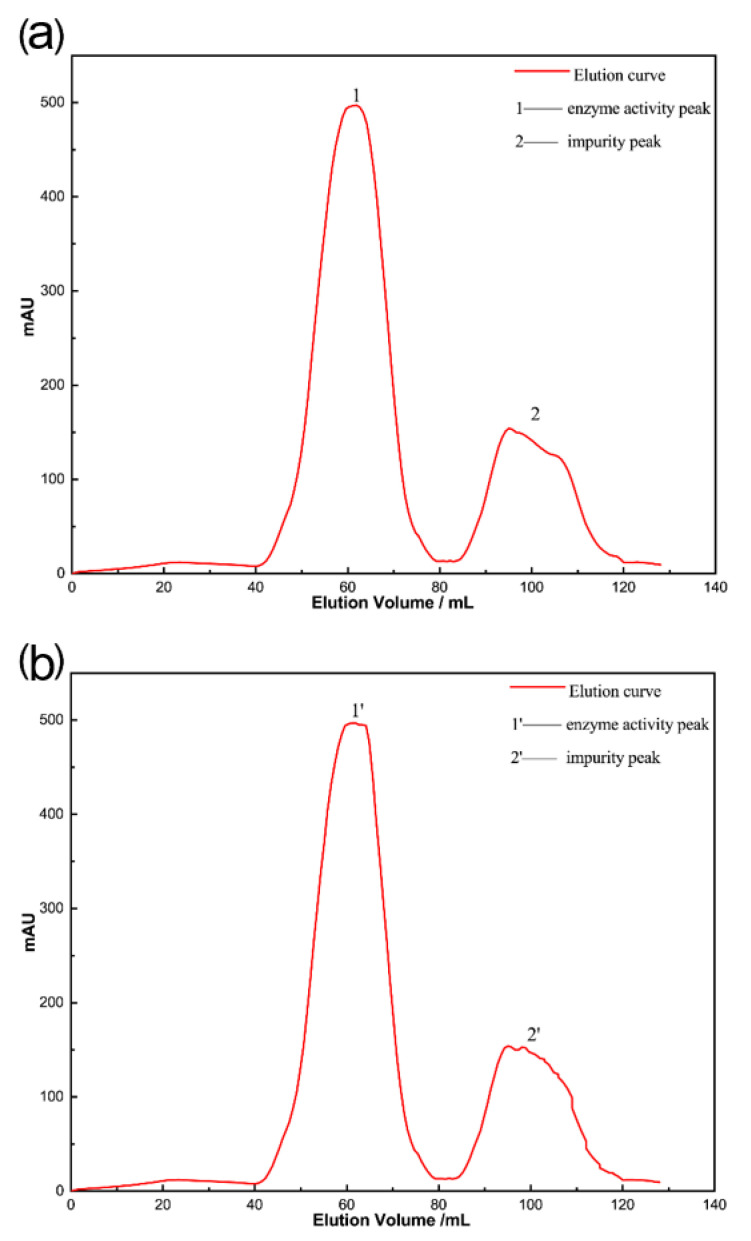
The gel chromatography spectra of modified enzyme: (**a**) caffeic acid modified enzyme (**b**) p-coumaric acid modified enzyme.

**Figure 2 molecules-28-00095-f002:**
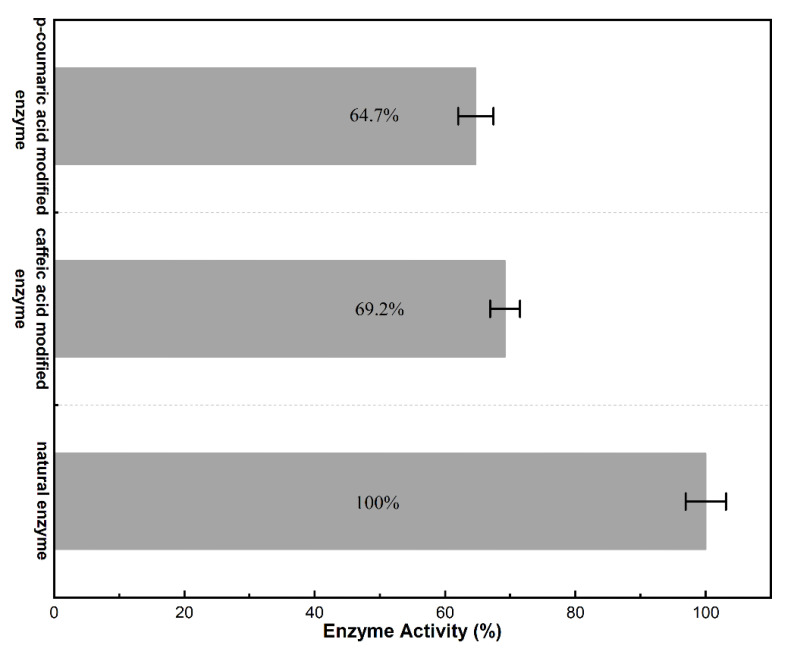
The relative enzyme activity of the natural and modified enzyme.

**Figure 3 molecules-28-00095-f003:**
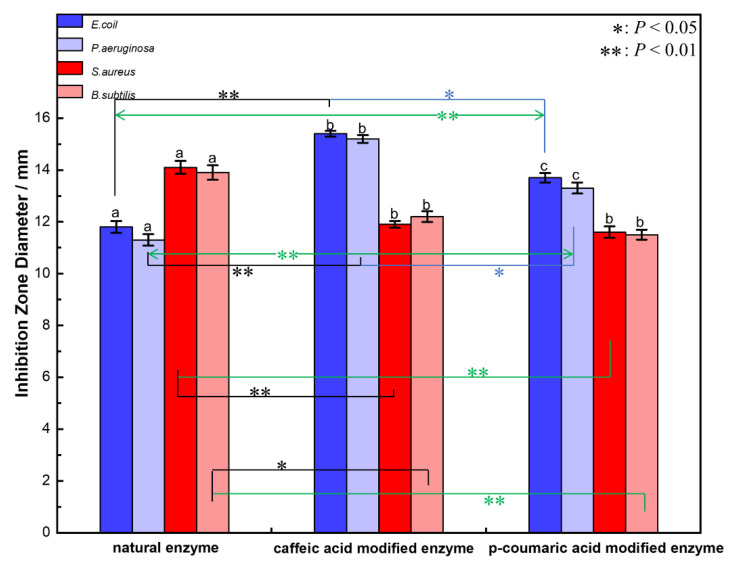
The bacteriostatic circle diameter of natural and modified enzyme samples to different bacteria (the values of different letters marked with the same strain were significantly different, *: *p* < 0.05, **: *p* < 0.01).

**Figure 4 molecules-28-00095-f004:**
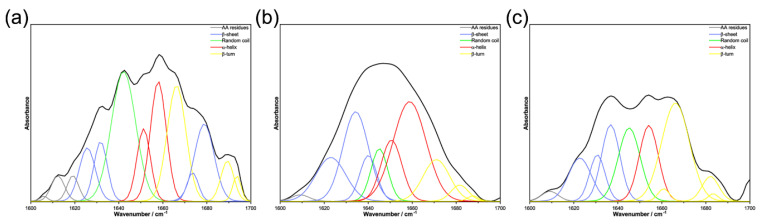
The amide I band fitting diagram of natural enzyme and modified enzyme: (**a**) natural enzyme, (**b**) caffeic acid modified enzyme, (**c**) p-coumaric acid modified enzyme.

**Figure 5 molecules-28-00095-f005:**
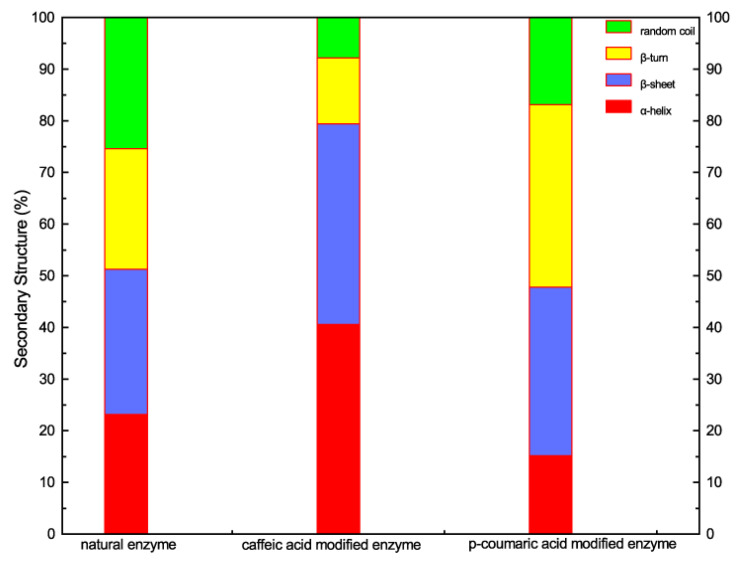
Visual secondary structure relative contents of natural enzyme and modified enzyme.

**Table 1 molecules-28-00095-t001:** Determination results of MIC of natural lysozyme and modified lysozyme on different test strains.

Bacteria	Sample	Blank	1.50 mg/mL	1.25 mg/mL	1.00 mg/mL	0.75 mg/mL	0.50 mg/mL	0.25 mg/mL
*Escherichia coli*	1	++	--	++	++	++	++	++
	2	++	--	--	--	--	--	++
	3	++	--	--	--	--	++	++
*Pseudomonas aeruginosa*	1	++	--	++	++	++	++	++
	2	++	--	--	--	--	--	++
	3	++	--	--	--	--	++	++
*Staphylococcus aureus*	1	++	--	--	--	--	++	++
	2	++	--	--	++	++	++	++
	3	++	--	--	++	++	++	++
*Bacillus subtilis*	1	++	--	--	--	--	++	++
	2	++	--	--	++	++	++	++
	3	++	--	--	++	++	++	++

1: natural enzyme; 2: caffeic acid modified enzyme; 3: p-coumaric acid modified enzyme; ++: bacterium growth; --: no bacterial growth.

**Table 2 molecules-28-00095-t002:** MIC of natural lysozyme and modified lysozyme to test strains.

Sample	*Escherichia coli*	*Pseudomonas aeruginosa*	*Staphylococcus aureus*	*Bacillus subtilis*
Natural enzyme	1.50 mg/mL	1.50 mg/mL	0.75 mg/ml	0.75 mg/mL
Caffeic acid modified enzyme	0.50 mg/mL	0.50 mg/mL	1.25 mg/mL	1.25 mg/mL
p-coumaric acid modified enzyme	0.75 mg/mL	0.75 mg/mL	1.25 mg/mL	1.25 mg/mL

**Table 3 molecules-28-00095-t003:** The distribution range of the secondary structure in D_2_O solution.

Secondary Structure	α-Helix	β-Sheet	β-Turn	Random Coil
Wavenumber * (cm^−1^)	1649–1658	1620–1640 & 1675–1680	1659–1674 & 1681–1696	1640–1648

* Data are from Susi and colleagues [18,19,20,21].

**Table 4 molecules-28-00095-t004:** The relative proportion of the four secondary structures.

		Secondary Structure (%)			
		α-Helix	β-Sheet	β-Turn	Random Coil
Sample	Natural enzyme	23.13 ± 0.08	28.10 ± 0.03	23.37 ± 0.02	25.41 ± 0.03
	Caffeic acid modified enzyme	40.53 ± 0.06	38.87 ± 0.02	12.78 ± 0.03	7.82 ± 0.01
	p-coumaric acid modified enzyme	15.17 ± 0.05	32.62 ± 0.08	35.33 ± 0.02	16.88 ± 0.01

**Table 5 molecules-28-00095-t005:** Comparison of modification effects of caffeic acid and p-coumaric acid modified lysozyme with other similar work.

Modifier	Lysozyme Source	Lytic Activity (The Lytic Activity of Native Lysozyme as 100%)	Antibacterial Effect after Modification	Ref.
Caffeic acid	Hen egg white	69.2%	The antibacterial impact against G− was greatly improved, whereas the efficacy against G+ was diminished to variable degrees and the total antibacterial spectrum was expanded.	This work
p-coumaric acid	Hen egg white	64.7%		This work
Dithiothreitol (DTT) and iodoacetamide	Hen egg white	59%		[6]
Hydrophobic pentapeptide	Hen egg white	55%		[7]
Palmitic acid	Hen egg white	62%		[8]
Myristic and stearic acids	Fresh hen egg white	60%, 65%		[9]
Perillaldehyde	Fresh hen egg white	72.3%		[10]
Cinnamic acid	Egg white	68.4%		[11]
Short and middle chain saturated fatty acids (caproic and capric acids)	Hen egg white	62%, 54%		[22]
Dextran	Hen egg white	13.3%		[23,24]

## Data Availability

Not applicable.

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
