# Peer review of "Study on Antibacterial Activity and Structure of Chemically Modified Lysozyme"

_molecules, 2022, doi:10.3390/molecules28010095_

Round 1
Reviewer 1 Report
The topic of the work is very interesting, fitting into the area of research on lysozyme, the aim of which is to use the potential, especially antibacterial, especially that obtained after modifying the enzyme. The manuscript presented for review is a work that fits into this trend, and presents the results of, in my opinion, preliminary research on this method of lysozyme modification, and I believe that the authors will extend this research to obtain preparations with much better properties. Despite the preliminary nature of the previous research, I believe that this work is worth publishing and I recommend it for publication, as it brings new elements to this area of research.
Detailed notes:
1. INTRODUCTION: Other, alternative methods of lysozyme modification should be mentioned, e.g. thermal, thermo-chemical or the recently presented method of modification using microwave radiation.
2. MATERIALS and METHODS: line 144 - imprecise information, what the statement means: "A certain amount of natural enzyme and modified enzyme were dissolved.."; this information should be clarified;
Lack of characteristics of native (purchased) lysozyme (What was the hydrolytic activity, molecular weight?), such information should be provided
3. RESULTS and DISCUSSION:
Fig. 1 and Fig. 2 - provide a legend for the title of the drawings describing peaks 1, 2 and 1' and 2'
Fig. 3 and Fig. 4 - provide statistical data (significant difference in the means at what level of significance p)
What was the value of hydrolytic activity for the obtained preparations of modified lysozyme?
Considerations on the structure of modified lysozyme should be supplemented (in further studies) with electrophoretic analysis, the results of which will show the fractional composition of the obtained preparations.
Lysozyme is an enzyme protein that plays a very important role in nature. Its main property is the destruction of Gram (+) bacteria. Its activity against Gram-negative bacteria is very limited. However, it has been proven that after the appropriate modification of the enzyme, it extends its antimicrobial activity, including its activity against Gram-negative bacteria, and many other valuable properties are revealed. Hence, a number of studies on lysozyme modifications are currently being carried out so that its great potential can be fully exposed and used. The work presented for review is part of this research trend and introduces some new elements. The authors have shown that within the acidic chemical modifications of lysozyme, organic acids can also be used, which have not been used for this purpose so far. This premise of the work is its original element. Although the effect of the modification is not yet fully satisfactory, the authors indicate the direction of research that can bring great benefits. Therefore, the basic conclusion of these studies is that organic acids can also be used for the chemical modification of lysozyme.
The presented research methodology includes the assessment of structural changes, which does not fully reflect the picture of changes that occur in the modified enzyme. At this stage of research, it can be accepted, but the research methodology must be supplemented with electrophoretic testing, which provides information about the phase composition of the enzyme. Native lysozyme occurs in the form of a monomer, while in the modified enzyme there are oligomers, and among them the most important is the dimer, which is responsible for the new properties of lysozyme. The electrophoretic test shows what oligomers are formed as a result of the modification. Using electrophoresis, the authors could answer the question of how lysozyme modifications with the use of organic acids affect the formation of the dimeric form of the enzyme.
Concerning the conclusions, references and comments to the drawings, I referred earlier in my review.
Author Response
Thank you for your valuable comments and contributions!Please see the attachment.

Reviewer 2 Report
The article Study on Antibacterial Activity and Structure of Chemically Modified Lysozyme shows the results of evaluating the effectiveness of the antibacterial ability of Egg white lysozyme modified with various organic acids. In general, the authors have done a lot of research aimed not only at characterizing the obtained modified compounds, but also at determining their antibacterial properties in comparison with natural enzymes. In general, this article corresponds to the subject of the declared Molecules journal, the article contains new data and can be accepted for publication after the authors answer a number of questions from the reviewer that arose during the analysis of this work.
1. In the abstract, the authors should reflect not only the results of the studies, but also the novelty of this work, as well as a number of distinctive features of the proposed modifications for Egg white lysozyme in order to increase the effectiveness of the antibacterial ability. The reasons for choosing Egg white lysozyme as an object of study should also be reflected, taking into account the fact that in some cases, as experiments have shown, the modification of Egg white lysozyme does not lead to a significant increase in efficiency in comparison with natural enzymes, as the authors report in the work.
2. When analyzing the data presented in Figures 1-2, the authors should present them on one slide to show the comparison and difference in the spectra.
3. The results presented in sections 3.3 and 3.4 require a more detailed analysis, since they reflect the most important points related to the modification of structures.
4. The authors should make a number of comparative data with other similar works, and not only with natural enzymes, to reflect the prospects of the compounds they propose.
5. From technical notes: the authors should indicate the manufacturers and brands of all chemicals and compounds used in the work, as well as equipment used for research.
6. Also, measurement errors must be added to all measured quantities.
Author Response

(The authors gave the same response as above.)

Reviewer 3 Report
1. -All the microbe's names should be in italics
2. Improve the abstract with your findings
3. What is the positive control used?
4. Authors did not present MIC values and not presented How MIC values are measured, please rewrite this.
5- Determination of the Inhibition Zone and its result is not compatible, in addition, no reference is cited about how to interpret it.
6. The mechanism of the antibacterial activity should be elaborately discussed in the results and discussion section.
7. Language used must be improved throughout the manuscript.
8. Authors must make sure and recheck that they followed the guidelines for the reference section as mentioned in the manuscript.
Author Response

(The authors gave the same response as above.)

Round 2
Reviewer 2 Report
The authors answered all the questions, the article can be accepted for publication.
Reviewer 3 Report
Thanks for the author's response.